# Role of HMGB1 in the Interplay between NETosis and Thrombosis in Ischemic Stroke: A Review

**DOI:** 10.3390/cells9081794

**Published:** 2020-07-28

**Authors:** Seung-Woo Kim, Ja-Kyeong Lee

**Affiliations:** 1Department of Biomedical Sciences, Inha University School of Medicine, Inchon 22212, Korea; swkim1@inha.ac.kr; 2Medical Research Center, Inha University School of Medicine, Inchon 22212, Korea; 3Department of Anatomy, Inha University School of Medicine, Inchon 22212, Korea

**Keywords:** HMGB1, DAMP, neutrophils, NETosis, stroke, platelets, thrombosis

## Abstract

Neutrophil extracellular traps (NETs) comprise decondensed chromatin, histones and neutrophil granular proteins and are involved in the response to infectious as well as non-infectious diseases. The prothrombotic activity of NETs has been reported in various thrombus-related diseases; this activity can be attributed to the fact that the NETs serve as a scaffold for cells and numerous coagulation factors and stimulate fibrin deposition. A crosstalk between NETs and thrombosis has been indicated to play a role in numerous thrombosis-related conditions including stroke. In cerebral ischemia, neutrophils are the first group of cells to infiltrate the damaged brain tissue, where they produce NETs in the brain parenchyma and within blood vessels, thereby aggravating inflammation. Increasing evidences suggest the connection between NETosis and thrombosis as a possible cause of “tPA resistance”, a problem encountered during the treatment of stroke patients. Several damage-associated molecular pattern molecules have been proven to induce NETosis and thrombosis, with high mobility group box 1 (HMGB1) playing a critical role. This review discusses NETosis and thrombosis and their crosstalk in various thrombosis-related diseases, focusing on the role of HMGB1 as a mediator in stroke. We also addresses the function of peptidylarginine deiminase 4 with respect to the interplay with HMGB1 in NET-induced thrombosis.

## 1. Introduction

Neutrophils infiltrate damaged brain tissue in the earlier stages of various pathologic conditions of the central nervous system (CNS) and produce proinflammatory cytokines, matrix metalloproteinases (MMPs), nitric oxide (NO), reactive oxygen species (ROS) and other cytotoxic molecules that accelerate brain damage [1,2]. Formation of neutrophil extracellular traps (NETs) has been recognized as an additional defense mechanism against infectious conditions [3]. NETs are webs of extracellular DNAs interspersed with histones and neutrophil granular proteins, such as myeloperoxidase (MPO) and neutrophil elastase (NE) [4]. Interestingly, although NETosis is a unique form of cell death characterized by membrane rupture (lytic form), vital NETosis (non-lytic form) releases NETs from the vesicle via nuclear budding without membrane breakdown [5,6]. Although NETs help in pathogen clearance during infections, NETs also play a critical role in various non-infectious diseases. However, excessive NET formation promotes inflammation and tissue damage in both conditions.

Stroke is the leading cause of disability and the third leading cause of mortality worldwide. However, currently, only two treatment regimens are approved for ischemic stroke by the U.S. Food and Drug Administration (FDA), pharmacological thrombolysis using tissue plasminogen activator (tPA) and mechanical removal of the thrombus via endovascular thrombectomy. Unfortunately, recanalization via tPA is achieved only in less than half of the recipients—even when the treatment starts within hours of symptom onset [7]. Although the exact reasons are unknown, analysis of increasing numbers of thrombi obtained from stroke patients after endovascular thrombectomy led us to infer that the thrombus composition may contribute to this so-called “tPA resistance” [8]. Histological and biochemical examination of those thrombi revealed the localization of neutrophils and citrullinated histone H3 (CitH3, a NETosis marker) in most thrombi [9,10,11] and demonstrated that ex vivo lysis of patient thrombi was more successful when DNase 1 was combined with standard tPA treatment [9]. Based on these reports, it can be suggested that NETs have a prothrombotic effect and confer resistance to fibrinolysis in acute ischemic stroke.

In cerebral ischemia, damage-associated molecular patterns (DAMPs) are massively released during the acute phase and mediate signaling for proinflammatory activation of microglia, astrocytes and endothelial cells. DAMPs also trigger excessive infiltration of the neutrophils into the damaged brain tissue [12]. Activated neutrophils produce ROS and proinflammatory cytokines and exhibit NETosis in brain parenchyma and within blood vessels, thereby aggravating inflammation and injury following ischemia [13,14]. Recently, we reported that that high mobility group box 1 (HMGB1), a well-studied DAMP molecule, induces NETosis in the ischemic brain through Toll-like receptor 4 (TLR4) and C–X–C motif chemokine receptor 4 (CXCR4) [15]. Recent review articles describes NETosis in cardiovascular diseases [16], DAMP and NETs in sepsis [17], HMGB1 in thrombus-related diseases [18] and NETs in thrombosis-related disease [19]. In this review, we discuss the role of DAMPs, especially HMGB1, in NETosis in cerebral ischemia, with a focus on the role of HMGB1 in NETosis-mediated thrombosis. We also summarize the role of peptidylarginine deiminase 4 (PAD4), an enzyme that condenses chromatin during the interplay between NETosis and thrombosis, theorizing a reciprocal modulation between HMGB1 and PAD4.

## 2. DAMPs in NET Formation in Non-Infectious Disease

Tissue injuries induce uncontrolled release of cellular components, some of which function as DAMPs [20]. DAMPs play physiological roles inside the cell, but when they are exposed to the extracellular milieu, they activate innate and adaptive immune responses [21]. Recent studies indicate that DAMPs such as HMGB1, histones and uric acid are important factors that trigger NET formation and exacerbate tissue injury in cases of sterile inflammation. DAMP-triggered NETosis was reported initially in autoimmune diseases. For example, crystals of monosodium urate (MSU) induce the development of chronic gout via the promotion of NET formation [22,23]. NETs-MSU crystal aggregates were observed in patients with systemic autoimmunity [23]. Released NETs components act as autoantigens for autoantibody production, so-called, anti-neutrophil cytoplasmic antibodies (ANCA), which cause autoimmune disease and further induce NETosis, thereby aggravating the disease [24]. Currently, an increasing number of reports reveal that diverse DAMP molecules induce and modulate NETosis in various non-infectious diseases (Table 1).

### 2.1. HMGB1

HMGB1, a well-known DAMP, is a non-histone chromosomal protein localized in the nucleus under normal physiological condition [25]. However, when it is present in the extracellular milieu after being released from the dying cells or after secretion from the activated immune cells, HMGB1 is involved in the induction and aggravation of inflammatory responses [26,27]. HMGB1 contains three conserved cysteine residues, C23, C45 and C106, and the functions of HMGB1 are determined by redox-state-dependent modifications of these cysteines and subsequent binding with their specific receptors. HMGB1 containing a disulfide bond between C23 and C45, termed disulfide HMGB1, binds to the TLR4 to induce cytokine production by macrophages [28]. Fully reduced HMGB1, termed all thiol HMGB1, contributes to chemoattractant formation by forming a hetero-complex with C–X–C motif chemokine ligand 12 (CXCL12), that synergistically promotes migration of immune cells after CXCR4 binding [29]. Fully oxidized HMGB1, termed sulfonyl HMGB1, does not bind to either TLR4 or CXCR4 and has no cytokine or chemokine activity [30]. The role of HMGB1 in NET formation has been reported in numerous non-infectious diseases. The first observation of colocalization of HMGB1 with NETs was reported in systemic lupus erythematosus (SLE), wherein HMGB1 was detected in complexes with nucleosomes and DNA and signified a potential marker for disease severity [31]. Additionally, localization of HMGB1 in association with extracellular DNA released from neutrophils was detected in the synovial cells of gout patients [2,3]. The first report on HMGB1-mediated induction of NETosis demonstrated that recombinant HMGB1 induced NET formation through interaction with TLR4 and that it was inhibited by the administration of neutralizing HMGB1 antibodies [32]. Huang et al. [33] showed that treatment of recombinant histone and HMGB1 elevated CitH3 levels and induced liver damage; however, these phenomena were suppressed in TLR4—Toll-like receptor 9 (TLR9)—and myeloid differentiation primary response 88 (MyD88)-deficient mice. Additionally, NETosis induced by tumor-infiltrating neutrophils promote proliferation, migration and invasion of glioma cells, wherein HMGB1 activates the NF-κB signaling pathway via receptor for advanced glycation end products (RAGE) and promotes IL-8 secretion in glioblastoma [34]. Interestingly, promotion of prothrombotic NET formation by administration of disulfide HMGB1 was observed in venous thrombosis, which was mediated by RAGE [35]. In our recent report, using the middle cerebral artery occlusion (MCAO) model, we demonstrated that HMGB1 acts as a NETosis inducer, wherein both, all thiol HMGB1 and disulfide HMGB1 are capable of inducing NETosis via CXCR4 and TLR4, respectively [15]. We also demonstrated that suppression of NETosis in ischemic brain by intranasal administration of PAD4 inhibitor ameliorated delayed inflammation and enhanced the repair process, which was evidenced by reduced immune cell infiltration and improved blood vessel formation [15].

### 2.2. Histones

In addition to HMGB1, histones released from damaged renal cells in acute kidney injury (AKI) triggered neutrophil recruitment and renal inflammation via interaction with TLR2 and TLR4, in turn activating MyD88, NF-κB and mitogen activated protein kinase (MAPK) signaling [36]. Similarly, in ischemic AKI, histones released from NETs enhanced tubular necrosis and remote organ injury [37]. During liver ischemia reperfusion (I/R), combined administration of recombinant histone and HMGB1 elevated CitH3 levels in the bone marrow-derived neutrophils, but enhanced CitH3 levels were suppressed in TLR4-, TLR9- and MyD88-deficient mice [33]. On the other hand, histones released after NETosis were also shown to promote glomerular necrosis [38] and be responsible for NET-mediated cytotoxicity in lipopolysaccharide (LPS)-induced acute lung injury [39]. Recently, accumulation of CitH3 and MPO-positive cells after administration of histones (H3 + H4) has been shown in human neutrophils, and induction of NET formation was observed in a histone dose-dependent manner [40]. 

### 2.3. ATP

Extracellular ATP released from dying cells also acts as a DAMP and triggers the activation of numerous immune and endothelial cells via P2X receptors. Previously, it has been reported that oxidized ATP (an ATP antagonist) suppressed MSU crystal-induced NET formation [41]. Additionally, inhibition of the P2X6 receptor resulted in the suppression of MSU crystal-induced NET formation [42], indicating the importance of purinergic receptors in NET formation. Recently, Sofoluwe et al. [43] have reported that ATP enhances phorbol 12-myristate 13-acetate (PMA)-induced NETosis, although ATP alone fails to induce NET formation in bone marrow-derived neutrophils. However, we found that ATP or BzATP (a prototypic P2X7R agonist) alone is capable of inducing the upregulation of PAD4 and CitH3 and subsequent NET formation in neutrophils isolated from circulating blood and also in the cortical penumbra of the ischemic brain following MCAO (manuscript in preparation) [44].

### 2.4. Monosodium Urate (Uric Acid)

Gout is a sterile inflammatory disease induced by MSU-deposition-driven leukocyte infiltration. MSU acts as a DAMP and induces NADPH oxidase (NOX)-dependent NET formation and accumulation [41], and during this process, macrophage-derived IL-1β promotes neutrophil recruitment and MSU-triggered NET production [45]. Contrastingly, treatment with EDTA, ATP antagonist, ROS scavenger, and DNase 1 significantly suppresses MSU-induced aggregation of NETs in human neutrophils [41]. Interestingly, MSU not only induces NETosis, but also induces the formation of extracellular traps (ETs) in eosinophils and basophils in an ROS-dependent manner [23,41]. During MSU-induced NETosis, P2Y6 and P2Y receptor blockers inhibit NET formation, indicating a pivotal role of purinergic signaling in MSU-mediated gout pathogenesis [42]. Unlike MSU, uric acid directly induces NET formation in the absence of NOX activation [46]. In systemic lupus erythematosus (SLE) patients, prolonged exposure to NETs and their released contents induces ANCA production [47], which in turn, induces NETosis and aggravates inflammation, leading to a vicious cycle of autoimmune inflammation [48]. NETs form stable complexes, thereby acquiring resistance against degradation by DNase 1, and resulting in aggravation of the autoimmune response in SLE patients [49]. Interestingly, HMGB1 promotes ANCA-inducing NET formation in conjunction with NOX via TLR2, TLR4 and RAGE during the pathogenesis of ANCA-associated vasculitis [50].

## 3. NETs and Thrombosis

Uncontrolled and excessive NET formation in the vasculature may contribute to pathologic thrombotic disorders [52]. NETs promote vessel occlusion by activating the platelets. NETs also provide a scaffold for platelets and red blood cells as well as for procoagulant molecules and stimulate coagulation by activating the intrinsic pathway and degrading the inhibitors of the extrinsic pathway [52,53].

### 3.1. NET Induces Thrombosis and Vice Versa

An association between NETs and thrombosis was first demonstrated by Fuchs et al. [53]. They showed that extracellular DNA extruded from NETs provided a scaffold for platelets and red blood cells and the high content of nucleic acids and histones rendered the NETs highly procoagulant by activating and aggregating platelets [53]. Activated platelets bound to procoagulant molecules, such as VWF (von Willebrand factor), fibronectin and fibrinogen, which were immobilized on NETs [53]. Additionally, the NET components promoted coagulation via the direct activation of procoagulant molecules or degradation of inhibitors of the tissue factor pathway. In deep vein thrombosis, NETs bind and activate factor XII, an inducer of the intrinsic coagulation pathway, thereby propagating the disease [54]. High expression of tissue factor (TF) in NETs and their association with thrombosis have been reported in vasculitis and sepsis [55,56], and especially the release of TF-bearing NETs at the site of inflammation results in the localized activation of the coagulation cascade [55]. Furthermore, proteinases such as elastase and cathepsin G from the released NETs degrade physiological coagulation inhibitors such as tissue factor pathway inhibitor (TFPI), thereby accelerating coagulation [57]. On the other hand, during the acute myocardial infarction, the interaction between thrombin-activated platelets with PMNs at the site of plaque rupture results in local NET generation and subsequent TF exposure [58]. Therefore, accumulating evidences indicate that NET induces thrombosis and vice versa.

Moreover, NET components such as, extracellular DNA, histones and nucleosomes have long been shown to be directly responsible for the prothrombotic effects of NET. Extracellular nucleic acid has been known to activate innate immune responses and induce coagulation after infection [59]. Significant elevations in plasma DNA concentrations and nucleosome levels were found in stroke patients and they were correlated with stroke severity [60,61]. In this respect, increased serum DNase 1 levels were reported in myocardial ischemia [62] and a polymorphism of DNase 1-associated less active DNase 1 was shown to be related with myocardial infarction [63], suggesting that high plasma DNA levels are indicative of this disease and that DNase 1 exerts a protective role. Significant elevations in nucleosome levels were also reported in the sera of stroke patients and these levels were correlated with the infarction volume and gross functional status [64]. Extracellular nucleosomes were also shown to contribute towards promoting coagulation and intravascular thrombus formation [57]. Increased levels of circulating nucleosomes and DNA were reported in a murine model of ischemic stroke, and neutralization of histones using an anti-histone antibody was found to have a protective effect in ischemic stroke mice, although a direct link to thrombosis was not demonstrated in this report [65]. Furthermore, histones are well known for their potency in direct thrombin generation, platelet activation and promoting coagulation [53,66,67,68]. Therefore, a coagulation platform provided by the concerted functions of histone–DNA backbone of NETs and stable fibrin scaffold in thrombi endow the thrombi with higher stability and rigidity, resulting in a prolonged clot lysis time [69].

### 3.2. NETs and Thrombosis in Ischemic Stroke

Presence of various NET components in the plasma of stroke patients has been proposed to be responsible for the prothrombotic effects of NETs and has been used to predict disease severity. Earlier, plasma DNA concentrations and nucleosome levels in patients with acute ischemic stroke have been shown to be correlated with stroke severity, with a potential to predict mortality and morbidity [60,61,64]. Recently, Valles et al. [70] reported that NETs were significantly abundant in the plasma of patients with acute ischemic stroke, and that the CitH3 levels at onset were independently associated with severity of atrial fibrillation (AF) and all-cause mortality at one year follow-up. More direct evidences regarding the NETs–thrombus relationship came from the histological studies performed on thrombi of patients with acute ischemic stroke that were retrieved during endovascular thrombectomy [9,10,11]. Laridan et al. [9] analyzed 68 thrombi retrieved from ischemic stroke patients and observed the colocalization of neutrophils and CitH3 and extracellular DNA in almost all thrombi. Interestingly, CitH3 levels were higher in cerebral thrombi of the cardioembolic origin than those with other etiologies. This observation was consistent with the results of the study analyzing thrombi retrieved from 145 acute ischemic stroke patients with large-vessel occlusion that revealed a high proportion of fibrin/platelets and high number of leukocytes in thrombus specimens of cardioembolic origin [11]. Laridan et al. [9] also found that older and more mature thrombi contain significantly more neutrophils and CitH3 compared to fresh thrombi, which is in agreement with the findings of Savchenko et al. [71] who observed NET formation predominantly during the organization phase in human venous thrombi. However, the abundance of NETs was, however, not correlated with the clinical outcome in a study that performed immunohistochemical analyses of thrombi from 108 acute ischemic stroke patients, in which thrombus content was correlated with the duration of endovascular therapy and number of devices used [10], suggesting that NETs may contribute to reperfusion resistance towards mechanical destruction.

Contribution of NETs to reperfusion resistance especially to enzymatic lysis was corroborated by numerous reports showing that combined treatment with DNase 1 and tPA showed significantly higher thrombi destruction than that in response to monotherapy in human cerebral thrombi [9,10]. The therapeutic potential of DNase 1 has also been supported by a study using animal models of transient MCAO, in which the administration of DNase 1 improved stroke outcome, which was determined by assessing the neurological and motoric behavior [65]. Interestingly, impaired DNase 1 activity is responsible for acute thrombotic microangiopathies and restoration of DNase 1 activity in the plasma by supplying recombinant human DNase 1 restored NET-degradation activity [72]. Therefore, NETs may contribute to reperfusion resistance against both, mechanical destruction and enzymatic lysis. As recombinant human DNase is clinically administered and is used as a safe pharmacological remedy to clear extracellular DNA in cystic fibrosis [73], it is intriguing to speculate that degradation of NETs by DNase may represent a promising supplement to tPA treatment in cerebral thrombolysis.

### 3.3. NETs and Thrombosis in Other Non-Infectious Diseases

In acute myocardial infarction, NETs were found in the coronary thrombi excised from patients [51,74,75]. Coronary thrombi of patients with acute myocardial infarction consist of neutrophils and NETs in close proximity to activated platelets and activated platelets commit neutrophils to NET generation [51]. NET contents in coronary thrombus were found to be associated with poor outcomes in myocardial infarctions and DNase activity is correlated negatively with infarct size, while recombinant DNase accelerated the lysis of coronary thrombi ex vivo [74]. Markers of NETs, for example double-stranded DNA levels, were also significantly associated with adverse clinical outcomes in coronary artery disease after 2 years [76]. Colocalization of CitH3 with NE [77] or with extracellular DNA-histone complexes [78] was also observed in intraluminal thrombi obtained from abdominal aortic aneurysm patients. Therefore, NETs were identified as a potential therapeutic target to improve the efficacy of tPA-induced thrombolysis in myocardial infarction.

In addition to cardiovascular diseases, NETosis has also been shown to promote thrombosis in many pathologic conditions. In transfusion-related acute lung injury (TRALI), the leading cause of death after transfusion therapy, NETs were detected in the lungs and plasma of TRALI patients and in the plasma of patients with acute lung injury [79]. In a mouse model of TRALI, activated platelets were found to induce NET formation and targeting platelet activation with either aspirin or tirofiban, an inhibitor of the platelet fibrinogen receptor, decreased NET formation and reduced lung injury [79]. Platelet-dependent NET formation has also been demonstrated in a mouse model of acute lung injury [80], wherein, platelet-induced NET formation requires stimulation of neutrophils by platelet chemokines that aggravated disease progress [80]. NET formation was also reported in a mouse model of I/R injury of the kidney, but treatment of the mice with the platelet inhibitor clopidogrel reduced NET formation and the tissue damage induced by I/R, demonstrating the critical role of platelets in NET formation [81]. Lethal thrombotic complications have been observed in a high proportion of cancer patients [82]. Cell-free DNA increases during the course of chemotherapy in cancer patients as well as in a mouse model of cancer, and cell-free DNA has been suggested as a novel procoagulant stimulus in cancer patients [83]. Mice bearing solid tumors develop accompanied neutrophilia during tumor progression, which is usually a sign of poor prognosis [84] and NETs accumulate in the peripheral circulation in mice with cancer [85]. Cancers predispose neutrophils to release extracellular DNA traps that contribute to cancer-associated thrombosis. Additionally, neutrophils from tumor-bearing mice were found to be more prone to form NETs than neutrophils from healthy mice [86]. Therefore, NETs have also been implicated in cancer-associated thrombosis and NETs were suggested as potential targets to prevent thrombosis in cancer patients.

## 4. HMGB1 in the Crosstalk between NETosis and Thrombosis in Cerebral Ischemia

Emerging evidences indicate that HMGB1 plays a critical role in thrombosis-related diseases [18]. HMGB1 induces NETosis and activates platelets; it is also released from activated platelets and extruded from NETosed neutrophils. Extracellular HMGB1 derived from activated platelets or NETosed neutrophils induces thrombosis and further aggravates NETosis (Table 2, Figure 1). Therefore, HMGB1 serves as a center of the crosstalk between NETosis and thrombosis and mediates the aggravation cascade resulting from this crosstalk. This chain of events may be responsible for causing not only cerebral ischemia, but also numerous diseases related to immunothrombosis.

### 4.1. HMGB1 Induces NETosis and Is Released after NETosis Following Cerebral Ischemia

As already mentioned in the previous paragraph, HMGB1 induces NETosis, for example, in pulmonary neutrophils purified from LPS-injected mice [32], in liver I/R animal model [33] and in vitro in static and in physiological flow conditions [51]. In our recent report, we demonstrated that HMGB1 induced NETosis in the ischemic brain, thereby exacerbating inflammation and hampering vascular recovery [15]. Interestingly, we found NETosis induction not only in the brain parenchyma, but also within blood vessels in the ischemic hemisphere soon after MCAO, and the induction of CitH3 in circulating neutrophils began prior to the migration to the brain parenchyma [15]. Massive accumulations of HMGB1 in stroke patients and stroke animal models have been reported by many research groups including ours. Since Goldstein et al. [87] first reported a significant elevation of serum HMGB1 level in patient with ischemic stroke within 24 h of symptom onset, elevation of serum HMGB1 levels in stroke patients has been reported in numerous papers, and it was correlated with the severity and functional outcome of the disease [88,89,90,91]. In experiments using murine stroke models, elevated HMGB1 levels were detected in the brain tissue [26,92,93,94,95,96,97,98], serum [26,98,99,100,101], plasma [15,92,102,103] and cerebrospinal fluid (CSF) [26,98,104,105]. Although in brain tissue, HMGB1 released from neurons after acute excitotoxic death may mainly be responsible for the postischemic HMGB1 elevation, in plasma, activated platelets also contribute to the HMGB1 elevation, as in case of deep vein thrombosis, leading to NETosis induction in the blood vessels [106]. Furthermore, it is important to emphasize that HMGB1 not only induces NETosis but is also extruded as a part of NETs. HMGB1 has been identified in NETs from patients with pediatric SLE [107], in extracellular DNA structures released from the synovial cells of patients with gout [22], in arterial coronary thrombi [51] and in conditioned media of PMNs after inducing NETosis [15,51]. Extruded HMGB1 in turn, contributes to further NETosis induction and platelet activation and subsequent aggravation of inflammation. Therefore, it appears that a vicious cycle exists between neuronal cell death, NETosis and thrombosis, with HMGB1 playing a critical role in mediating the detrimental effects exerted by this cycle (Figure 1).

### 4.2. Activated Platelets Release HMGB1

The activation of platelets may serve as a trigger for so-called, immunothrombosis, and recent studies have shed light on the importance of HMGB1 in this context. However, it was intriguing to find HMGB1 expression in human platelets [108], because platelets are anucleate, whereas HMGB1 is a well-known nuclear protein. In the resting state of platelets, HMGB1 is localized in the cytoplasm, however, it is translocated to the surface after activation [108]. A close link between the platelet activation and HMGB1 release has been reported in numerous papers, in which HMGB1 was released from platelets following treatment with thrombin, collagen, ADP or CRP [51,109,110,111]. Moreover, HMGB1 expression on the surface of circulating platelets was markedly upregulated in patients with systemic sclerosis [112,113], trauma [114] and deep vein thrombosis [106]. This activated platelet-derived extracellular HMGB1 exerts prothrombotic function (Table 2). Interestingly, thrombin and collagen increase HMGB1 levels in exosomes derived from human platelets [111]. In this study, it was found that HMGB1 levels in these exosomes were higher in subjects older than 65 years and lower in the aspirin-treated group, implying that cargo proteins in human plasma platelet-derived exosomes may be used as biomarkers for platelet abnormalities [111].

### 4.3. Activated Platelet-Derived HMGB1 Exhibits Prothrombotic Function

HMGB1 released from activated platelets has been shown to promote microvascular thrombosis, resulting in excessive fibrin deposition in the glomeruli, prolonged plasma clotting times and increased 15 day-mortality in thrombin-induced rat model of disseminated intravascular coagulation [115]. In this animal model, HMGB1 also stimulated TF expression on monocytes and inhibited the anticoagulant protein C pathway mediated by the thrombin–thrombomodulin complex [115]. Ahrens et al. [110] reported that HMGB1 was highly expressed in platelet-rich human coronary artery thrombi and bound to activated platelets via RAGE, contributing to proinflammatory functions. In trauma patients, HMGB1 on the surface of the circulating platelets was found to be markedly upregulated [114]. Interestingly, mice lacking HMGB1 in platelets revealed that platelets are the major source of HMGB1 within thrombi, therefore these mice exhibited reduced thrombus formation and platelet aggregation, resulting in increased bleeding times and organ damage during experimental trauma/hemorrhagic shock [114]. Activated platelet-derived HMGB1 interferes with the myocardial repair process by suppressing recruitment of mesenchymal stem cells through a downregulating hepatocyte growth factor receptor MET via a TLR4-dependent process [109]. Activated platelet-derived HMGB1 also induces autophagy and promotes NET generation in a RAGE-dependent manner [51]. With respect to the underlying signaling, while TLR4-MyD88 signaling is involved in HMGB1 binding to platelets for impairing myocardial repair [109] and in trauma [114], the importance of RAGE and not TLR4 has been reported in coronary artery disease and in NET [51,110]. The differences in the signaling pathways between these reports cannot be explained at present, and the different pathologic conditions may be the main reason.

### 4.4. Activated Platelet-Derived HMGB1 Induces NETosis

In addition to the platelet-derived HMGB1-mediated platelet activation and prothrombotic effect, platelet-derived HMGB1 also induces NETosis. Maugeri et al. [51]. Reported that activated platelets present HMGB1 to neutrophils and commit them to NET generation; however, the same was not observed in the presence of competitive antagonists of the HMGB1 or while using HMGB1^−/−^ platelets. Using a transgenic mouse specifically lacking HMGB1 in platelets and megakaryocytes, Dyer et al. [106] also demonstrated that the prothrombotic effect of platelet-derived HMGB1 is mediated through NET formation, and specifically via the release of extracellular DNA during NET formation [106]. Therefore, HMGB1-mediated NETosis induction may be one of the mechanisms underlying the prothrombotic effect of platelet-derived HMGB1.

## 5. PAD4 in the Crosstalk between NETosis and Thrombosis in Cerebral Ischemia

NETosis is initiated via PAD4-mediated citrullination of histones [116]. PAD4 is a member of the PAD enzyme family that citrullinates protein substrates and modulates various cellular processes, including gene expression, differentiation and inflammatory responses [117]. Protein citrullination by PAD4 converts the positively charged arginine residue to the neutrally charged citrulline (e.g., in histone H3 and H4 that results in chromatin decondensation). During the NETosis, PAD4 decondensed chromatin and is released into the extracellular environment along with decondensed chromatin [118]. Notably, PAD4 remains active in the extracellular environment, where it can citrullinate other substrates including plasma proteins [118]. PAD4 has been found in the plasma of patients with various diseases and dysregulated PAD activity and abnormal levels of protein citrullination are associated with the progression of those diseases. In fact, increased concentrations of extracellular PAD4 have been found in the plasma of patients with rheumatoid arthritis (RA) [119], multiple sclerosis [120], Alzheimer’s disease [121], sepsis [122] and malignant tumors [123]. Additionally, mitigation of disease severity due to PAD4 deficiency or by treatment with PAD inhibitors was reported in mouse models of RA [123,124], hypoxic ischemic insult in neonates [125], cardiac dysfunction [126] and neurodegenerative disorders [120]. It is important to note that PAD4 deficient mice not only failed to produce citrullinated histones in thrombi but were also unable to form stable thrombi in deep vein thrombosis, suggesting the importance of PAD4 in NET-thrombosis interaction [127]. In the next section, function of PAD4 in cerebral ischemia, which is associated with NETosis–thrombosis interaction, will be addressed with special focus on the relationship with HMGB1.

### 5.1. PAD4 Induction Cerebral Ischemia

Importance of PAD4 in cerebral ischemia was initially reported in a mouse model of hypoxic ischemic insult in neonates, in which inhibition of PAD4 using a pan-PAD inhibitor Cl–amidine reduced microglial activation, cell death and infarct volume [125]. Presently, upregulation of PAD4 has not been reported in stroke patients and direct experimental substantiation of the causal relationship between PAD4 and stroke severity obtained from PAD4 mutant is lacking. However, recently, Kang et al. [128] reported significant induction of PAD4 in the peri–infarct cortex 3 days after surgery in a mouse model of focal cortical cerebral ischemia generated using electrocoagulation of distal MCA. In addition, we also found a significant induction of PAD4 in cortical penumbra of ischemic hemisphere at 12–24 h post MCAO and in blood neutrophils purified from the rat MCAO model, peaking at 12 h post-MCAO (submitted) [44]. Protective effect of PAD4 inhibition in cerebral ischemia has been reported, wherein intranasal administration of Cl–amidine suppressed delayed immune cell infiltration and increased vessel formation in a rat MCAO model, with no reduction of infarct volume [15]. These results were corroborated by a recent report showing that disruption of NETs by DNase 1 and inhibition of NET formation by genetic ablation of PAD4 or Cl–amidine treatment increased neovascularization and vascular repair and improved functional recovery [128]. These results indicate that NETosis in cerebral ischemia aggravates delayed inflammation and impairs recovery processes, and induction of PAD4 plays a critical role.

### 5.2. Reciprocal Modulation between PAD4 and HMGB1 in Cerebral Ischemia

It can be theorized that a reciprocal modulation exists between PAD4 and HMGB1. In terms of the function of HMGB1, it upregulates PAD4 expression in neutrophils. Here, we showed that PAD4 expression was significantly induced in neutrophils isolated from peripheral blood after treating cells with recombinant HMGB1 for 2 h (Figure 2A), which is slightly earlier than the CitH3 inductions reported in our previous report (Figure 2D; [15]). Interestingly, both disulfide- and all thiol HMGB1 can induce PAD4 in a TLR4- and CXCR4-dependent manner, respectively (Figure 2B,C), similar to the CitH3 inductions observed in our previous report (Figure 2E,F; [15]). Additionally, in MCAO model, treatment with neutralizing HMGB1 antibody or antagonistic peptide for HMGB1 (HMGB1 A box) at 3 h post-MCAO suppressed CitH3 induction and NET formation detected at 12 h post-MCAO in both cerebral cortex and circulating blood neutrophils isolated from MCAO animals [15]. On the other hand, in case of the function of PAD4, PAD4 citrullinates HMGB1. PAD4 citrullinates HMGB1 at arginine residues localized in the inter-domain region between A and B boxes; this modification enables calpain to digest HMGB1 into two fragments, leading to efficient chromatin decondensation [129]. In the absence of PAD4, calpain alone did not induce chromatin decondensation, whereas the concerted action of calpain and PAD4 enhanced it, suggesting a novel mechanistic role of PAD4 in NET formation [129]. As HMGB1 not only induce NETosis but is also a part of the extruded NETs [15], reciprocal regulation between HMGB1 and PAD4 may contribute to this aggravation cycle in cerebral ischemia. Other functions of citrullinated HMGB1 which may be relevant to NETosis and thrombosis need further study.

### 5.3. Direct Prothrombotic Effect of PAD4 by Citrullination of Coagulation-Related Proteins

Most of the earlier studies using PAD4^−/−^ mice demonstrated the prothrombotic function of PAD4 associated with its NETosis-inducing effect [71,127]. However, PAD4 also exerts direct prothrombotic effects via the citrullination of plasma proteins involved in the coagulation process. Abnormal expression or activation of PAD4 in RA synovium was known to be responsible for high levels of citrullinated antithrombin in the plasma of RA patients, leading to aggravated RA pathology [124]. Additionally, increased PAD4 expression has been observed in the blood and tissues of patients with malignant tumors, and these elevated PAD4 levels were significantly associated with high thrombin activity caused by citrullinated antithrombin [123]. Citrullination of fibrinogen has also been found in the plasma of patients with RA [130]. Recently, Sorvillo et al. [131] reported that PAD4 citrullinates a disintegrin and metalloproteinase with thrombospondin type-1 motif-13 (ADAMTS13), which reduces its activity. Since the von Willebrand Factor (VWF) is secreted from activated endothelial cells and binds platelets, forming VWF–platelet strings, and these strings are cleared from the vessel wall by ADAMTS13 [132], inhibition of ADAMTS13 activity by PAD4 is an important prothrombotic mechanism, leading to the accumulation of VWF–platelet strings on blood vessel walls. Importance of ADAMTS13 has been reported in stroke patients, wherein the VWF/ADAMTS13 ratio was a predictive of outcome [133] and associated with an increased risk of ischemic stroke [134]. Furthermore, in animal models of stroke, ADAMTS13 not only affects infarct volume [135], but also controls neuroinflammation [92] and recovery [136], along with the maintenance of cerebrovascular integrity [137]. Thus, citrullination of ADAMTS13 is a direct and novel mechanism for the prothrombotic function of PAD4.

## 6. Conclusions

The connection between NETosis and thrombosis is recognized in various thrombosis-related diseases including stroke. Various studies indicate that extracellular HMGB1 plays a central role in the induction of NETosis, activation of platelets, propagation of NETosis and subsequent thrombosis. HMGB1 activates neutrophils and platelets and is also released from both these cells as a result of these activations. Therefore, HMGB1 appears to play a pivotal role in the crosstalk between NETosis and thrombosis and mediates the aggravation cascade arising as a result of this crosstalk. This chain of events may be responsible for numerous diseases related to immunothrombosis; however, particularly in the case of cerebral ischemia, the interplay between NETs and thrombosis may contribute towards reperfusion resistance to both mechanical destruction and enzymatic lysis. Therefore, a clear grasp of HMGB1, NETs, and their roles in thrombosis in cerebral ischemia is important to devise strategies not only for ameliorating delayed inflammation and impaired recovery processes, but also for preventing secondary thrombi formation or even recurrent stroke.

## Figures and Tables

**Figure 1 cells-09-01794-f001:**
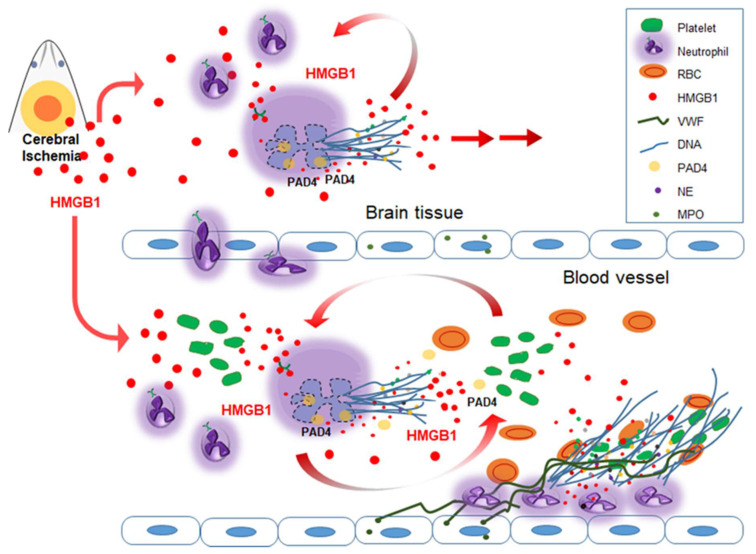
Diagram of the interplay between NETosis and thrombosis in cerebral ischemia and the function of HMGB1 in this process. HMGB1 is released from neurons and glia during the acute phase of cerebral ischemia. HMGB1 recruits neutrophils to the damaged brain region. NETosis occurs both in brain parenchyma and inside the blood vessel. In the brain parenchyma, extruded NET components induce brain damage and HMGB1 included in expelled neutrophil extracellular traps (NETs) accelerates this process. Inside the blood vessel, activated platelets release HMGB1, which in turn further induces NETosis and activates various procoagulant molecules. Platelets adhere to the endothelium by binding to von Willebrand factor (VWF) and interact with NETs that trap circulating procoagulant factors and red blood cell (RBC). HMGB1 derived from activated platelets and NETosed neutrophils mediates the interactions between neutrophils and platelets, promoting thrombus formation. PAD4 initiates chromatin decondensation and also citrullinates plasma proteins involved in coagulation process. HMGB1—high mobility group box 1; NE—neutrophil elastase; MPO—myeloperoxidase; VWF—von Willebrand factor.

**Figure 2 cells-09-01794-f002:**
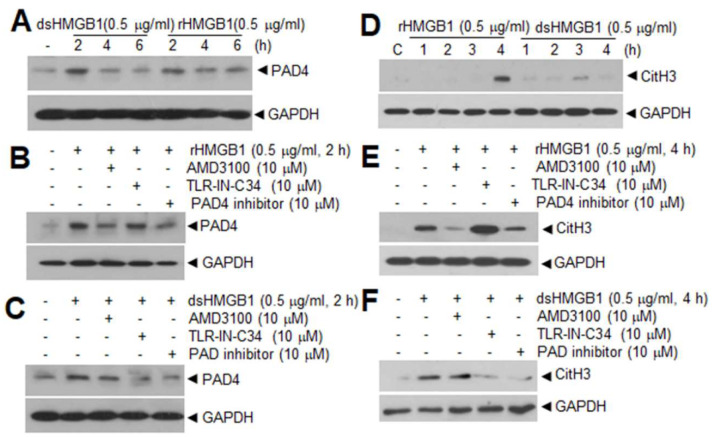
Both all-thiol HMGB1 and disulfide HMGB1 increases PAD4 and CitH3 levels in peripheral neutrophils isolated from the blood. (**A**) PAD4 and (**D**) CitH3 levels were examined by immunoblotting after treating neutrophils isolated from peripheral blood with all-thiol HMGB1 or disulfide HMGB1 (0.5 µg/mL) for indicating durations; (**B**,**C**,**E**,**F**) Blood PMNs were pretreated with AMD3100 (10 µM, CXCR3 antagonist), TLR–IN–C34 (10 µM, TLR4 antagonist) or Cl–amidine (10 µM, a PAD4 inhibitor) for 30 min and then treated with all-thiol HMGB1 (0.5 µg/mL) or disulfide HMGB1 (0.5 µg/mL) for 2 h for PAD4 (**B**,**C**) or 4 h for CitH3 (**E**,**F**). PAD4 and CitH3 levels were subsequently assessed by immunoblotting. Data in A–C are unpublished ones and in D–F were retrieved from our previous study (Kim et al., 2019) [15].

**Table 1 cells-09-01794-t001:** Damage-associated molecular patterns (DAMPs) inducing NETosis in non-infectious disease.

Inducer	Diseases	Signaling	Ref.
HMGB1	MCAO	TLR4, CXCR4	[15]
HMGB1 Histones	Liver I/R	TLR4, TLR9, MyD88	[33]
HMGB1	AMI	RAGE	[51]
HMGB1	DVT	RAGE, MyD88	[35]
HMGB1	Lung Injury	TLR4	[32]
MSU Uric acid	Gout	NADPH Oxidase NF-κB	[41,46]
ATP	MCAO Gout	PKC, NADPH Oxidase NADPH Oxidase	[41,44]

I/R—ischemia and reperfusion; AMI—acute myocardial infarction; DVT—deep vein thrombosis; MSU—monosodium urate; TLR—toll like receptor; RAGE—receptor for advanced glycation endproducts; CXCR—C–X–C motif chemokine receptor; MyD88—myeloid differentiation primary response 88.

**Table 2 cells-09-01794-t002:** HMGB1 in the interplay between NETosis–thrombosis.

HMGB1 Action Point	Diseases (Model System)	Receptor (Signaling)	Results Observed	Ref.
NETosis induction by HMGB1	LPS-injected mice (Lung)	TLR4	Recombinant HMGB1 induced NET and it was inhibited by the administration of neutralizing HMGB1 antibodies	[32]
	Liver I/R	TLR4, TLR9	Recombinant histone and HMGB1 elevated CitH3 levels and induced liver damage	[33]
	AMI	RAGE	HMGB1 induced NET formation, however, HMGB1^−^/^−^ platelets failed to elicit NETs	[51]
	MCAO	TLR4, CXCR4	Recombinant disulfide or all thiol HMGB1 induced NETosis and NETosis was inhibited by anti-HMGB1 antibody or HMGB1 A box	[15]
HMGB1 released from NETosed neutrophils	Pediatric SLE		HMGB1 was identified in NETs from pediatric SLE patients	[107]
	Gout		HMGB1 was detected in extracellular DNA	[22]
	AMI		HMGB1 was detected in NETs in arterial coronary thrombi	[51]
	Co-culture of neutrophils		Blocking HMGB1 in extruded NETs suppressed NETosed-neutrophil induced neuronal cell death	[15]
HMGB1 release from activated platelets	Exosomes		Thrombin and collagen increased HMGB1 levels in exosomes derived from human platelets	[111]
	Trauma		Marked upregulation of HMGB1 in platelets	[114]
	AMI	RAGE	HMGB1 was released from activated circulating platelets	[51]
	DVT		Platelets accounted for most circulating HMGB1	[106]
	Coronary artery thrombi	RAGE	HMGB 1 was in platelet-rich human coronary artery thrombi	[110]
	Systemic sclerosis		HMGB1 was released from activated of circulating platelets	[112,113]
Thrombosis induction by platelets-derived HMGB1	DIC		Activated platelet-derived HMGB1 promotes microvascular thrombosis and stimulated TF expression	[115]
	Coronary artery thrombi	RAGE	HMGB 1 released from platelets in human coronary artery thrombi activated platelets	[110]
	Trauma	TLR	Mice lacking HMGB1 in platelets exhibited reduced thrombus formation and platelet aggregation, resulting in increased bleeding times and organ damage	[114]
	Myocyte apoptosis	TLR	Activated platelet-derived HMGB1 interfered myocytes survival by suppressing mesenchymal stem cell recruitment	[109]
	AMI	RAGE NETosis	Activated platelet-derived HMGB1 promoted NET generation, leading to thrombo-inflammatory lesions	[51]
	DVT	NETosis	Platelet-derived HMGB1 enhanced neutrophil recruitment and NET formation and promoted DVT formation	[106]

I/R—ischemia and reperfusion; AMI—acute myocardial infarction; DVT—deep vein thrombosis; SLE—systemic lupus erythematosus; DIC—disseminated intravascular coagulation; TLR—toll like receptor; RAGE—receptor for advanced glycation endproducts; CXCR—C–X–C motif chemokine receptor.

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
