# Peer review of "Role of HMGB1 in the Interplay between NETosis and Thrombosis in Ischemic Stroke: A Review"

_cells, 2020, doi:10.3390/cells9081794_

Round 1
Reviewer 1 Report
The review well explained the innate immune system after ischemic brain injury. Brain tissue injuries induce the uncontrolled release of cellular components; damage-associated molecular patterns (DAMPs) such as HMGB1 which is one of the important factors that trigger neutrophil extracellular traps. The manuscript showed how the cerebral thrombosis could be r-TPA resistant. The mechanism could be the future medical treatment strategies for ameliorating inflammation and preventing secondary thrombi in stroke patients.
Excellent writing.
Author Response
In the revised manuscript, we clearly cited unpublished data as “(manuscript in preparation) [44]” and included it in the Reference section as reference number [44].
Reviewer 2 Report
This review discuss the crosstalk between NETosis and thrombosis along with the role of damage-associated molecular patterns (DAMPs) in neutrophil extracellular trap (NET) formation. It also addressed the complex relationship between HMGB1 and peptidylarginine deiminase 4 (PAD4), an enzyme that condenses chromatin during thrombosis. The main emphasis of the discussion was to understand the interplay between NETosis and thrombosis in cerebral ischemia and the function of HMGB1 in this process.
Understanding the intricate relationship between these processes and their contribution to thrombosis in cerebral ischemia is of great importance. It will helpful in getting an insight into the reasons of “tPA resistance”, a major challenge associate with recanalization via tPA in stroke patients. The manuscript is well written and organized. The concept are well discussed and informative. The data is current as majority of the studies discussed in the manuscript are not more than 5 year old. I have few minor suggestions
- Table 1 is not discussed in the text.
- There are grammatical and typographical mistakes in the manuscript. Few sentences are very confusing and difficult to follow.
- Although the title says “Role of HMGB1 in the interplay between NETosis and thrombosis in ischemic stroke,” most of the studies discussed are not of stroke models.
Author Response
- Table 1 is not discussed in the text.
Response: We mentioned Table 1 in the revised manuscript.
- There are grammatical and typographical mistakes in the manuscript. Few sentences are very confusing and difficult to follow.
Response: We read carefully our manuscript and corrected mistakes throughout the manuscript.
Although the title says “Role of HMGB1 in the interplay between NETosis and thrombosis in ischemic stroke,” most of the studies discussed are not of stroke models.
Response: We understand reviewer's point. Although, not many reports on stroke are published at this moment, available data clearly indicate the importance of the role of HMGB1 in NETosis and thrombosis, and related investigations are going on and will be reported soon.